# Gender Differences in the Medical Treatment of Peripheral Artery Disease

**DOI:** 10.3390/jcm10132855

**Published:** 2021-06-28

**Authors:** Damien Lanéelle, Gabriella Sauvet, Jérôme Guillaumat, Jean Eudes Trihan, Guillaume Mahé

**Affiliations:** 1Vascular Medicine Unit, Centre Hospitalier Universitaire de Caen Normandie, 14000 Caen, France; laneelle-d@chu-caen.fr (D.L.); guillaumat-j@chu-caen.fr (J.G.); 2COMETE Laboratory, INSERM 1075, Université de Caen, 14000 Caen, France; 3Medical School, Université de Versailles Saint-Quentin-en-Yvelines, 78180 Montigny-Le-Bretonneux, France; gabriellasauvet@gmail.com; 4Vascular Medicine Unit, Centre Hospitalier Universitaire de Poitiers, 86021 Poitiers, France; jean-eudes.trihan@chu-poitiers.fr; 5Vascular Medicine Unit, Centre Hospitalier Universitaire de Rennes, 35033 Rennes, France; 6INSERM CIC 1414, Université de Rennes, 35033 Rennes, France

**Keywords:** peripheral arterial disease, optimal medical treatment, gender differences

## Abstract

Background/Objectives: Peripheral arterial disease is a frequent and severe disease with high cardiovascular morbidity and mortality. However, female patients appear to be undertreated. Objectives: The primary goal was to compare the prescription of optimal medical treatment (OMT) of peripheral arterial disease between women and men in primary health care. Material and methods: An observational retrospective study was based on the data collected from general practitioners (GP) office in Brittany. Results: The study included 100 patients, aged 71 ± 10 years old, with 24% of women. Compared to men, women received the OMT less frequently (29.2% vs. 53.9%, *p* = 0.038), especially after 75 years old. Antiplatelet therapy was largely prescribed (100%), statins less frequently (70.8% women vs. 85.5% men), and prescription of renin-angiotensin-aldosterone system inhibitors was still not optimal in the two genders (41.7% women vs. 61.9% men). Active smoking is important for both women and men (33% and 30% respectively). Conclusion: Optimal medical treatment of peripheral artery disease is insufficiently prescribed, especially in women in this region of France.

## 1. Introduction

Lower extremity peripheral arterial disease (PAD) is a frequent and increasing pathology, estimated at 235 million people affected worldwide and associated with substantial morbidity and mortality [1]. Lifestyle change and optimal medical treatment are the core of the treatment [2,3,4]. Antiplatelet aggregation inhibitor (AAP), a statin, and an antihypertensive drug (Angiotensin-Converting Enzyme inhibitor, ACE, or an angiotensin Receptor Blocker, ARBs) is the Optimal Medical Treatment (OMT) for PAD [5].

PAD in women is less frequent than in age-matched men; however, women are more likely to have atypical symptoms and greater walking impairment with progressive functional decline [6,7,8,9,10]. Research into gender differences in the prescription of medical and lifestyle therapies for PAD is lacking but women are treated less intensively to achieve cardiovascular risk factor targets [11] and appear to be treated less favorably for PAD than men at discharge from a tertiary-care teaching hospital [12]. Scientific statements have increasingly promoted the need for increased awareness and further research into gender-specific concerns in PAD. Women have been consistently underrepresented in clinical trials [6] and, accordingly, conclusions are hard to draw from case reports.

The present study examined the effect of gender on treatment medical management in patients with PAD in primary health care. The main objective was to compare the proportion of women with the OMT to men.

## 2. Materials and Methods

A retrospective observational study was conducted using data collected anonymously from general practitioners (GP) office in Brittany after a declaration of conformity and registration at the National Institute of Health Data (MR 0711090620). Thirty-two GPs were randomly selected out of 5288 (0.6%) in Brittany to search for the last five consecutive patients with PAD, based on medical record, who have consulted in the last 12 months and with a collection of medical elements on the whole medical record. Each patient was notified by a written information sheet and, in the absence of opposition, data were collected from the medical record. The study was conducted in Brittany (Brittany contains 4 departments) in France from June to August 2020 and involved 7 to 9 GP per department. The general collected data were age; sex; weight and height; history of vascular diseases (coronary heart disease (CHD), cerebrovascular disease (CVD), define by transient ischemic attack, stroke and/or carotid endarterectomy); smoking (current, at least 1-year cessation or never); diabetes; hypertension (HT), dyslipidemia, low-density lipoprotein cholesterol (LDL-c) and creatinemia. The PAD data collected were Leriche and Fontaine classification: stage 1 (asymptomatic), stage 2 (claudication), stage 3–4 (chronic permanent ischemia without and with trophic disorder, respectively); history of revascularization procedures; ankle–brachial index (ABI); date to the last vascular or cardiologist follow-up and all current medical and surgical treatments. OMT was defined as the association of Antiplatelet agent (or anticoagulation) + Statin + Angiotensin ACE/ARBs.

Statistical analyses: Quantitative results are expressed as mean ± standard deviation and analyzed by the Student’s test after verification of the conditions of application, including the normal distribution of observations. Qualitative results are expressed in frequency and percentage, which have been compared by the χ^2^ test or the Fisher exact test if necessary. The analyses, tables and figures were performed on RStudio (version 1.3.959) and edited on Microsoft Excel 2016. A value of *p* < 0.05 was chosen as statistically significant in the two-tailed analysis.

## 3. Results

### 3.1. Characteristics of the Patients with Lower Extremity Peripheral Artery Disease

A total of 100 patients with PAD were included (Table 1). The study population was composed by 72% of patients over 65 years of age (75% female and 71% male) and 28% were over 75 years of age. With respect to the Leriche and Fontaine classification, 54% were stage 2 (21% without revascularization, 34% with revascularization), 27% were stage 1 (8% without revascularization, 19% with revascularization), and 17% had a stage 3–4 PAD (3% without revascularization, 14% with revascularization). The mean LDL level was 1.03 ± 0.35 g/L with 50% LDL < 1 g/L, 16% with LDL < 0.7 g/L and 5% with LDL < 0.55 g/L.

Forty-eight percent of patients received OMT, regardless of the severity of the disease or risk factors and 42% for patients with only a PAD (patients with a history of CHD and CVD excluded).

### 3.2. Gender Differences

There were no significant gender differences with respect to risk factors or according to the severity of the disease but the proportion of women with the OMT (AAP or anticoagulant + statin + ACE or ARBs) was significantly lower than that of men (29% vs. 54%, *p* = 0.038, Table 2), particularly in the oldest patients ≥75 years old (*p* = 0.016).

There was no difference in the medical treatment according to the gender of the GP. On statins, the LDL target is mostly not reached with 80% of patients with LDL > 0.7 g/L. In hypertensive patients (*n* = 65), 61% had OMT compared to only 23% in non-hypertensive patients (*p* < 0.001). Adverse side effects were mainly myalgia (8%) for statins and cough (4%) for ACE or ARBs, there was no gender difference in side effects.

A follow-up visit to a vascular physician occurred in 90% of patients within 2 years (87% for women and 91% for men), 9% within 2–5 years and 1% beyond. ABI was not performed at the last vascular consultation in 25 patients (in 18 revascularized and 7 non-revascularized patients). A check-up with a cardiologist was performed within 2 years in 75% of patients (70.8% women and 76.3% men), 13% within 2–5 years and 12% beyond.

## 4. Discussion

Fifty percent of patients in the study received OMT, regardless of the severity of the disease but solely 42% for patients with only PAD (excluding patients with a history of CHD and CVD). This is an improvement over the 2003 situation observed in the French ATTEST study using a similar population in primary care (31% of the patients received OMT and 13% of patients with only a PAD, excluding CHD and CVD, out of 3811 patients, recruited from 3020 GPs). This improvement can be explained by the increase in the quality of follow-up, especially for patients with PAD alone, possibly related to the professionalization of vascular medicine in a primary care setting [13]. This improvement mainly concerns male patients. Indeed, this study shows that the proportion of women with PAD (29.2%) who received the optimal medical treatment (antiplatelet or anticoagulant therapy, statin and ACE or ARBs) is lower than that of men (53.9%), especially after age 75. This gender difference has already been found in a larger scale Canadian observational study of PAD in 2010 from a tertiary-care teaching hospital (*n* = 5962 patients; 18.2% women vs. 22.4% men; *p* < 0.001).

The study was not designed to determine whether gender difference in OMT had an impact on morbidity. The clinical impact of the gender difference in OMT remains to be studied.

PAD prevalence among women may differ within elderly age groups. As such, a study of demographic data from PAD patients evaluated at a vascular laboratory revealed that women were 3.3 years older than men and made up a larger percentage of the patients aged >65 years (*n* = 410, 69% women vs. 57% men) [14]. The same trend is observed in the sample presented here, without reaching the significance level, probably because of the smaller sample size (women were 5 years older than men; *p* = 0.059). A trend already reported in an Italian study in 2008 (*n* = 231; 2.7 years; *p* = 0.058) [15].

Antiplatelet therapy was widely prescribed (100%, counting those on anticoagulants), statin prescription (82%) appears to be improving over previous studies (62.2% from the REACH study, *n* = 5861) in both men (85.5%) and women (70.8%) but the prescription of ACE/ARBs was still suboptimal in both genders (61.9% and 41.7%, men and women respectively).

Non-prescription of statin (28%) appears to be only partially related to the adverse effect (7.8%) and non-prescription of ACE/ARBs (43%) is clearly not secondary to the onset of cough or hyperkalemia (3.7%). The fact could be partially explained by the discrepancies in the latest recommendations, some recommending the prescription of ACE/ARBs for all patients with PAD and others only in case of hypertension [2,3,4,5].

Statins have proven to be effective for patient with PAD in reducing major vascular events (strokes, revascularizations, major coronary events) both in registry studies such as REACH [1] and in randomized studies such as the HPS study [2], regardless of the initial LDL level. AAP also been shown to be effective in reducing major vascular events, notably clopidogrel [3] in patients with PAD. ACE and ARBs also allow a reduction of major vascular events in patients with PAD, regardless of hypertension. In this PAD population, the HOPE study compared the efficacy of ramipril 10 mg vs. placebo in 3577 diabetics and found a significant reduction in major vascular events of 25% in symptomatic patients with PAD [4]. Similarly, the ONTARGET study compared telmisartan 80 mg, ramipril 10 or both, with 1150 patients with PAD in each group and found an identical decrease in major vascular events, with no benefit to the combination [5].

Standardization of international recommendations regarding drug management could probably lead to improved prescribing, particularly with regard to renin-angiotensin-aldosterone system inhibitors.

Smoking is still significant in patients with PAD (31% actives smokers) and smoking cessation remains stable (51%) in comparison with 2003 data from the REACH study (50.9%, *n* = 5861 patients with PAD).

Another interesting observation is the rate of non-achievement of ABI measurement (25%) at the last consultation with the vascular physician, as part of the annual follow-up for severe disease (72% with a history of revascularization among those who did not have an ABI measurement). This can be explained by the duration of this measure for the vascular physician, its price set by the French health agency (€21), and the imprecision of the measure in some patients. However, performing ABI at each PAD appointment is recommended by international guidelines [2,5].

It is also interesting to note that the percentage of revascularization is particularly high (68%). This is not surprising given the rates of endovascular and surgical revascularization of patients with PAD, which exceed 10% per year in France [16], whereas the data presented here concerns the history of revascularization without time limit.

Among the remaining questions to be clarified was the clinical impact of differential treatment by gender. Anecdotally, non-statin cholesterol-lowering drug was not screen. The most important limitation of this study is its sample size of 100 patients with PAD, which seems low in relation to the incidence of this disease. This is related to the lack of external funding for this study and the need for on-site data collection from each randomized GP. Other limitations are those inherent to retrospective analyses such as selection bias and the presence of unmeasured confounders even if the randomization of GPs tends to limit it. Moreover, this was a Breton population and extrapolation of these results is not possible.

## 5. Conclusions

Optimal medical treatment of peripheral artery disease is insufficiently prescribed, especially in women in this region of France.

## Figures and Tables

**Table 1 jcm-10-02855-t001:** Baseline characteristics of the 100 patients with lower extremity peripheral artery disease.

Variables	Value (*n* = 100)
Female sex	24%
Age, years (mean ± SD)	71 ± 10
Body-mass index	
≥30 kg/m^2^	21%
25 to 30 kg/m^2^	38%
Body-mass index, kg/m^2^ (mean ± SD)	27 ± 5
Risk factors	
Diabetes	26%
Hypertension	65%
Low-density lipoprotein cholesterol, g/L (mean ± SD)	1.03 ± 0.35
Active smoking	31%
Past smoking ^†^	51%
History of vascular disease	
Coronary Heart Disease	32%
Cerebrovascular Disease	14%
Chronic kidney Disease Stage	
Stage 4 (GFR < 30 mL/min/1.73 m²)	1%
Stage 3 (GFR 30 to 59 mL/min/1.73 m²)	18%
Stage 2 (GFR 60 to 89 mL/min/1.73 m²)	44%
Stage 1 (GFR > 89 mL/min/1.73 m²)	37%
GFR, mL/min/1.73 m² (mean ± SD)	80.8 ± 21.8
ABI	
≤0.90	51%
0.91 to 0.99	5%
1.00 to 1.40	19%
No data available	25%
ABI, no unit (mean ± SD)	0.79 ± 0.22
PAD Leriche and Fontaine Classification	
Stage 3–4	17%
Stage 2	54%
Stage 1	27%
Surgical treatment	
Revascularization	68%
Amputation	9%
Medical treatment	
Antiplatelet agent or anticoagulation	100%
Statin	82%
ACE/ARBs	57%
OMT ^‡^	48%

SD: Standard Deviation; GFR: Glomerular Filtration Rate; PAD: Peripheral Artery Disease; ABI: Ankle-Brachial Index; ACE: Angiotensin-Converting Enzyme Inhibitor; ARBs: Angiotensin Receptor Blockers. ^†^ Patients were considered to be past smokers if they achieved at least a 1-year cessation. ^‡^ OMT: Optimal Medical Treatment means antiplatelet agent or anticoagulation + Statin + ACE/ARBs.

**Table 2 jcm-10-02855-t002:** Gender differences in risk factors, presentation and management of peripheral vascular disease.

Variable	Women (*n* = 24)	Men (*n* = 76)
Age, years (mean ± SD)	74 ± 10	69 ± 10
Body-mass index		
≥30 kg/m^2^	17%	22%
25 to 30 kg/m^2^	33%	40%
Body-mass index, kg/m^2^ (mean ± SD)	25 ± 4	27 ± 5
Risk factors		
Diabetes	33%	24%
Hypertension	75%	62%
LDLc, g/L (mean ± SD)	1.09 ± 0.36	1.01 ± 0.34
Active smoking	33%	30%
Past smoking ^†^	37%	55%
History of vascular disease		
Coronary Heart Disease	29%	32%
Cerebrovascular Disease	4%	17%
Chronic kidney Disease Stage		
Stage 4 (GFR < 30 mL/min/1.73 m²)	0%	1%
Stage 3 (GFR 30 to 59 mL/min/1.73 m²)	21%	17%
Stage 2 (GFR 60 to 89 mL/min/1.73 m²)	46%	43%
Stage 1 (GFR > 89 mL/min/1.73 m²)	33%	39%
GFR, mL/min/1.73 m² (mean ± SD)	78.04 ± 18.8	81.68 ± 22.7
ABI		
≤0.90	65%	69%
0.91 to 0.99	6%	7%
1.00 to 1.40	29%	24%
Not measured	24%	29%
ABI, no unit (mean ± SD)	0.83 ± 0.19	0.78 ± 0.22
PAD Leriche and Fontaine Classification		
Stage 3–4	17%	17%
Stage 2	52%	56%
Stage 1	30%	37%
Surgical treatment		
Revascularization	67%	68%
Amputation	4%	10%
Medical treatment		
Antiplatelet agent or anticoagulation	100%	100%
Statin	71%	85%
ACE/ARBs	42%	62%
OMT ^‡^	29% *	54% *

SD: Standard Deviation; LDLc: Low-density lipoprotein cholesterol; GFR: Glomerular Filtration Rate; PAD: lower extremity peripheral arterial disease; ABI: Ankle-Brachial Index; ACE: Angiotensin-Converting Enzyme Inhibitor; ARBs: Angiotensin Receptor Blockers. ^†^ Patients were considered to be past smokers if they achieved at least a 1-year cessation. ^‡^ OMT: Optimal Medical Treatment means antiplatelet agent or anticoagulation + Statin + ACE/ARBs. *: *p* < 0.05.

## Data Availability

The raw data supporting the conclusions of this article will be made available by the authors, without undue reservation.

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
