# Peer review of "Gender Differences in the Medical Treatment of Peripheral Artery Disease"

_jcm, 2021, doi:10.3390/jcm10132855_

Round 1

Reviewer 1 Report

An interesting paper regarding differences in optimal medical treatment of peripheral arterial disease between women and men. However, before further considerations the following issues should be addressed.

Differences between gender themselves have no clinical significance. It would be more interesting to know if the difference found by the authors regarding OMT had any impact on follow-up outcomes, mortality and/or morbidity (rate of percutanenous interventions/open surgeries, including amputations) during follow-up period.

Process of recruitment to study is unclear (random/consecutive), consecutive, selected.

I have doubts regarding the most appropriate assessment of renal function. Where did you adopted such classification from? Chronic kidney disease (CKD) classification of NICE (National Institute for Health and Care Excellence) is more commonly applied.

As for retrospective and multicenter (here in many GPs’ outpatient clinics) study, a number of 100 patients is relatively small. This fact should be pointed much stronger in a subsection of ‘Discussion’ dealing with study limitations

Conclusions are not supported by the findings of your study. You did not compare past medical therapy with the current status. In my opinion you should change conclusions.

Author Response

Comments and Suggestions for Authors by reviewer 1

  1. An interesting paper regarding differences in optimal medical treatment of peripheral arterial disease between women and men. However, before further considerations the following issues should be addressed.
  2. Differences between gender themselves have no clinical significance. It would be more interesting to know if the difference found by the authors regarding OMT had any impact on follow-up outcomes, mortality and/or morbidity (rate of percutanenous interventions/open surgeries, including amputations) during follow-up period.
  3. Process of recruitment to study is unclear (random/consecutive), consecutive, selected.
  4. I have doubts regarding the most appropriate assessment of renal function. Where did you adopted such classification from? Chronic kidney disease (CKD) classification of NICE (National Institute for Health and Care Excellence) is more commonly applied.
  5. As for retrospective and multicenter (here in many GPs’ outpatient clinics) study, a number of 100 patients is relatively small. This fact should be pointed much stronger in a subsection of ‘Discussion’ dealing with study limitations
  6. Conclusions are not supported by the findings of your study. You did not compare past medical therapy with the current status. In my opinion you should change conclusions.

Authors’ responses to the reviewer

  1. Thank you for your careful review of this manuscript, we have addressed the details of the revisions and responses to your comments point by point.
  2. This is a very relevant and interesting remark. However, we cannot perform this analysis because it is a cross-sectional study without any follow-up. However, given the number of patients, the aim of this study was not to discuss the relevance of international recommendations on the treatment of PAD but to observe a possible difference in prescription related to gender. We have modified the manuscript to highlight this element: “The study was not designed to investigate whether any gender difference regarding OMT had an impact on follow-up outcomes, mortality and/or vascular morbidity during a follow-up period so we cannot conclude whether this gender difference in medical treatment has a clinical impact. This remains to be studied.” p5, line 131.
  3. Thank you for this comment, which will allow us to improve the quality of the manuscript. The selection of GP clinics was randomized but patients in each clinic were included consecutively. We have improved the presentation of the recruitment process in the method section as follows: “We randomly selected 32 GP out of 5288 (0.6%) in Brittany to search for the last five consecutive patients with a PAD” p2, line 51.
  4. NICE Chronic kidney disease (CKD) classification was used in this study and we have modified table 1 and table 2 accordingly.
  5. Thank you for this comment, we are aware of the limitations and we have highlighted this aspect in the discussion in the following way; “The most important limitation of this study is its sample size of 100 patients with PAD, which seems low in relation to the incidence of this disease. This study was a pilot study in one region of France. Larger studies are required to confirm this result.” p6, line 92.
  6. We apologize and agree with the reviewer that the conclusion does not reflect our results. We have included this comparison with past medical treatments in the discussion and modified the conclusion and abstract as follows; “Optimal medical treatment of peripheral artery disease is insufficiently prescribed, especially in women in this region of France.”

Reviewer 2 Report

The abstract should be formatted to conform the style of the journal. 

Please reword the second sentence to However, female patients appear to be under-treated. 

Introduction--Line 24 please revise "individual studies" to case reports

Table 2. The percentage incidence are useful. I believe that the manuscript would be strengthened by generating mean and standard deviations for many of the parameters with an additional statistical analysis for parameters such as creatine clearance. 

the conclusion in the abstract is that renin angiotensin inhibitors should be considered for treatment, but the real data set is about gender differences. Please revise the abstract to reflect the data. 

the data sets also do not correlate clinical treatments with clinical outcomes, but lump all of the data together and expect the reader to dissect out any salient features. It would be better to more clearly present the relationships between the clinical data and clinical outcomes. 

Please refrain from the use of personal pronouns in scientific writing. 

Author Response

Comments and Suggestions for Authors by reviewer 2

  1. The abstract should be formatted to conform the style of the journal.
  2. Please reword the second sentence to However, female patients appear to be under-treated. Introduction--Line 24 please revise "individual studies" to case reports
  3. Table 2. The percentage incidence are useful. I believe that the manuscript would be strengthened by generating mean and standard deviations for many of the parameters with an additional statistical analysis for parameters such as creatine clearance.
  4. the conclusion in the abstract is that renin angiotensin inhibitors should be considered for treatment, but the real data set is about gender differences. Please revise the abstract to reflect the data.
  5. the data sets also do not correlate clinical treatments with clinical outcomes, but lump all of the data together and expect the reader to dissect out any salient features. It would be better to more clearly present the relationships between the clinical data and clinical outcomes.
  6. Please refrain from the use of personal pronouns in scientific writing.

Authors’ responses to the reviewer

  1. We have corrected the abstract to conform the style of the journal.
  2. Thank you for this suggestion, the sentence has been modified accordingly.
  3. Thank you for this comment, which will allow us to improve the quality of the manuscript. We have improved the presentation of the results by adding the mean and standard deviation of GFR values, ABI, BMI and by using the NICE Chronic Kidney Disease classification for tables 1 and 2. Note that we performed an additional statistical analysis on the mean value of ABI, BMI and creatinine clearance according to gender (not significant).
  4. We apologize and agree with the reviewer that the conclusion does not reflect our results. We have included the discussion about renin angiotensin inhibitors in the discussion and modified the conclusion as follows; “Optimal medical treatment of peripheral artery disease is insufficiently prescribed, especially in women.”
  5. This is a very relevant and interesting remark. However, we cannot perform this analysis because it is a cross-sectional study without any follow-up. However, given the number of patients, the aim of this study was not to investigate clinical outcomes according to medical treatment but to observe a possible difference in prescription related to gender. We have modified the manuscript to highlight this element: “The study was not designed to investigate whether gender difference regarding OMT had an impact on follow-up outcomes, mortality and/or vascular morbidity during follow-up period so we cannot conclude whether this gender difference in medical treatment has a clinical impact. This remains to be studied.” p5, line 45.
  6. Indeed, we used the term "we" 8 times and “our” 5 times. We have changed the wording of the sentences in question to make them impersonal.

Round 2

Reviewer 1 Report

Dear Authors,

Thank you for your corrections. In my opinion your paper sounds much better after revision.

I would recommend to accept for publication.

Author Response

Dear reviewer,

Thank you for your review and expertise.

Reviewer 2 Report

Table 2--are the differences between men and women statistically significant? 

Please reword the following passage to state the rationale behind the study. 

The reference to "this" is also not clear.

The study was not designed to investigate whether gender difference regarding OMT 136
had an impact on follow-up outcomes, mortality and/or vascular morbidity during fol- 137
low-up period so it is therefore not possible to conclude whether this gender difference in 138
medical treatment has a clinical impact. This remains to be studied. 139
PAD prevalence among women may differ within elderly age groupsThe study was not designed to investigate whether gender difference regarding OMT 136
had an impact on follow-up outcomes, mortality and/or vascular morbidity during fol- 137
low-up period so it is therefore not possible to conclude whether this gender difference in 138
medical treatment has a clinical impact. This remains to be studied. 139
PAD prevalence among women may differ within elderly age groupsThe study was not designed to investigate whether gender difference regarding OMT 136
had an impact on follow-up outcomes, mortality and/or vascular morbidity during fol- 137
low-up period so it is therefore not possible to conclude whether this gender difference in 138
medical treatment has a clinical impact. This remains to be studied. 139
PAD prevalence among women may differ within elderly age groups

Author Response

Dear reviewer,

Please find enclosed corrected manuscript. 

Comments and Suggestions for Authors:
1 Table 2--are the differences between men and women statistically significant? 
2 Please reword the following passage to state the rationale behind the study (lines 136-139). 
The reference to "this" is also not clear.

Authors’ responses to the reviewer
1.    Thank you for your review. The differences between man and women in Table 2 are not statistically significant, except for the OMT, as indicated by the * on the line in question. We have corrected by highlighting this in the legend.

2.     We have modified this sentence as follows: The study was not designed to determine whether gender difference in OMT had an impact on morbidity. The clinical impact of the gender difference in OMT remains to be studied.